# When band convergence is not beneficial for thermoelectrics

Junsoo Park [1✉], Maxwell Dylla[2], Yi Xia[2], Max Wood[2], G. Jeffrey Snyder[2✉] & Anubhav Jain [1✉]

Band convergence is considered a clear benefit to thermoelectric performance because it increases the charge carrier concentration for a given Fermi level, which typically enhances charge conductivity while preserving the Seebeck coefficient. However, this advantage hinges on the assumption that interband scattering of carriers is weak or insignificant. With first-principles treatment of electron-phonon scattering in the $CaMg_2Sb_2$-$CaZn_2Sb_2$ Zintl system and full Heusler $Sr_2SbAu$, we demonstrate that the benefit of band convergence can be intrinsically negated by interband scattering depending on the manner in which bands converge. In the Zintl alloy, band convergence does not improve weighted mobility or the density-of-states effective mass. We trace the underlying reason to the fact that the bands converge at a one k-point, which induces strong interband scattering of both the deformation-potential and the polar-optical kinds. The case contrasts with band convergence at distant k-points (as in the full Heusler), which better preserves the single-band scattering behavior thereby successfully leading to improved performance. Therefore, we suggest that band convergence as thermoelectric design principle is best suited to cases in which it occurs at distant k-points.

---

[1] Energy Technologies Area, Lawrence Berkeley National Laboratory, Berkeley, CA, USA. [2] Department of Materials Science and Engineering, Northwestern University, Evanston, IL, USA. ✉email: qkwnstn@gmail.com; jeff.snyder@northwestern.edu; ajain@lbl.gov

Thermoelectrics represent a clean energy technology for power generation from waste heat and refrigerant-free cooling. A thermoelectric material's performance is described by its figure of merit, $zT = \frac{\alpha^2 \sigma}{\kappa_{lat} + \kappa_e} T$, where $\alpha^2 \sigma$ is the thermoelectric power factor (PF), composed of the Seebeck coefficient ($\alpha$) and electrical conductivity ($\sigma$). The total thermal conductivity is the sum of electronic thermal conductivity and lattices thermal conductivity ($\kappa_{lat} + \kappa_e$). A more direct metric for assessing a material's potential thermoelectric performance with optimized doping is the quality factor[1-4], given by $\beta \propto \frac{\mu_w}{\kappa_{lat}}$. Here, $\mu_w = \mu m_D^{\frac{3}{2}}$ is known as weighted mobility, composed of mobility ($\mu$) and the density-of-states (DOS) effective mass ($m_D$). The goal of designing high-$zT$ thermoelectric materials rests on two main objectives: enhancing the electronic performance, represented by $\mu_w$ or the PF, and reducing lattice heat dissipation represented by $\kappa_{lat}$.

Band convergence is one of the most successful approaches for systematic improvement of electronic performance[5]. In essence, when multiple bands align in energy, they generate a higher carrier concentration ($n$) for a given Fermi level ($E_F$). This usually translates to increased $\sigma$ for given $\alpha$ enhancing the PF, or increased $m_D$ for given $\mu$ enhancing $\mu_w$. Beginning with PbTe[6], band convergence has been used to interpret the improved performance in many well-known thermoelectric compounds and alloys[7-18]. In a similar spirit, materials with inherent band multiplicity and degeneracy have been sought after for high intrinsic thermoelectric performance[19-28]. As such, a variety of descriptors and metrics for good thermoelectric materials universally promote a high multiplicity of Fermi surface pockets ($N_v$)[1,2,29,30]. For example, weighted mobility is also often written as $\mu_w = \mu N_v^{\frac{2}{3}} m_d$ where $m_d$ is the single-pocket DOS effective mass.

Yet, such a direct correlation of $N_v$ with improved thermoelectric performance neglects the implicit dependence of $\mu$ and $m_D$ on $N_v$ through interband (or intervalley) scattering. Experimental and theoretical analyses of multi-valley PbTe and Si suggest that although there is a definite increase in $\mu_w$ due to pocket multiplicity, it is perhaps half of that expected purely from the number of pockets[1]. Other demonstrations have emerged that scattering considerations complicate the benefit of $N_v$[21,31-33]. Modeling based on parabolic bands has recently proposed that, under interband scattering, the convergence of bands is guaranteed to be beneficial only if a lighter band converges with a heavy band[34]. A related concern is the relative location of band pockets in the Brillouin zone. Band pockets that are distant in reciprocal space are typically subject to weaker scattering overall[1,35] and, in certain cases, parity restrictions may further weaken intervalley transitions between distant symmetry-degenerate pockets[36]. In contrast, when multiple distinct bands are at the same k-point, there is little reason to presume that interband scattering would not possibly negate the benefit of band convergence.

In some experimental reports, the benefit of band convergence appears limited. Notably, in CaMg$_2$Sb$_2$, the Sb $p_z$ state lies above the Sb $p_x$ state at the valence band maximum (VBM) at the $\Gamma$-point, whereas in CaZn$_2$Sb$_2$, Sb $p_x$ lies above Sb $p_z$ to form the VBM at $\Gamma$. The $p_z$-state is split from $p_x$ and $p_y$ by crystal field, and the latter two are split by spin-orbit coupling (SOC). However, when CaZn$_{2-x}$Mg$_x$Sb$_2$ solid solutions were synthesized, $\mu_w$ did not peak at the expected point of full-band convergence at $x = 0.86$[37]. Other than for the Mg-end ($x = 2$), both the PF and $\mu_w$ essentially plateaued for all $x$ all the way to the Zn-end ($x = 0$). Alloy disorder scattering of carriers was suggested to be one culprit, but it alone cannot account for it since band convergence is often achieved through alloying. This warrants investigations at a more fundamental level of the scattering behaviors and thermoelectric response with respect to band convergence.

We herein investigate if the lack of performance increase could be attributable to an intrinsic process, namely electron-phonon (e-ph) scattering, including deformation-potential scattering (DPS) and polar-optical scattering (POS). We perform first-principles e-ph scattering and transport computations on CaZn$_{2-x}$Mg$_x$Sb$_2$ Zintl alloys as well as several hypothetical modifications of this band structure. Our results indicate that interband e-ph scattering and its changing behavior with band convergence are indeed inherently responsible and sufficient to explain the stagnant PF and $\mu_w$ in the Zintl alloys. We also highlight the critical role played by the relative locations of bands within a Brillouin zone: band convergence or multiplicity at distant k-points is more promising than it occurring at the same k-point, by design. In emphasis, this study strictly concerns the electronic performance represented by the PF and $\mu_w$, not $zT$ which may also benefit from reduced $\kappa_{lat}$ via alloying for band convergence. Also, we interpret "band convergence" strictly as shifts in band energies, i.e., the closing off energy offset between two bands of interest, independent of how they may also change in shapes during alloying in experiments. Note that where convenient and unambiguous, we use the Hartree atomic units ($\hbar = e = m_e = 4\pi\epsilon_0 = 1$).

## Results

We begin by theoretically confirming the occurrence of band convergence in CaZn$_{2-x}$Mg$_x$Sb$_2$ solid solutions. We calculate density-functional theory (DFT)[38] electronic structures including SOC using Vienna ab initio simulation package (VASP)[39-42] with projector-augmented wave (PAW) pseudopotential[43] and Perdew-Burke-Ernzerhof (PBE) exchange-correlation functional[44]. We perform the calculations on the midway alloy, CaZnMgSb$_2$, as well as CaMg$_2$Sb$_2$, an end compound (as explained in Supplementary Fig. S2, the CaZn$_2$Sb$_2$ endpoint is omitted). As seen in Fig. 1a, b, the two valence bands essentially converge in energy for the 1:1 ratio of Zn and Mg, closing their 0.096 eV offset (hereafter rounded to 0.1 eV) in CaMg$_2$Sb$_2$. These results suggest that the bands likely did converge in the experiments for the alloy (albeit at $x = 0.86$), but still failed to translate to higher thermoelectric performance. Of note, the upper band is heavier than the lower band in CaZnMgSb$_2$, and it is the light lower band that rises to converge on the heavy upper band as $x$ decreases. The upper band has a band effective mass (inverse curvature) $m \approx 0.4$, and the lower band has $m \approx 0.1$.

To probe the effect of band convergence, we compare the performance of the converged alloy with respect to the CaMg$_2$Sb$_2$ endpoint as well as several hypothetical band structures. As summarized in Fig. 1c, they include the CaZnMgSb$_2$ with a band offset enforced to be the same value as that of CaMg$_2$Sb$_2$ (but retaining the band shapes of CaZnMgSb$_2$); CaZnMgSb$_2$ with no band offset, i.e., fully converged bands (nearly the case from DFT with no adjustment); CaZnMgSb$_2$ with the light band practically removed (offset by 1 eV and rendered irrelevant); and CaZnMgSb$_2$ with the heavy band practically removed (offset by 1 eV and rendered irrelevant).

Figure 2a, b demonstrates that, whether measured by $\mu_w$ or the peak PF, the band-converged configuration is not the best design target, even with a lighter band converging onto a heavier band. The fully converged scenario is far outperformed by the hypothetical light-band-only case, matches the performance with 0.1 eV offset, and manages to slightly outperform the hypothetical heavy-band-only case. The light-band-only case is most desirable owing to its highest $\mu$ by far, as seen in Fig. 2c. The most interesting comparison is between the full band-converged case and the two bands offset by 0.1 eV. The former yields higher $\mu$ due to the convergence of the lighter band, but lower $\alpha$ and $m_D$, as

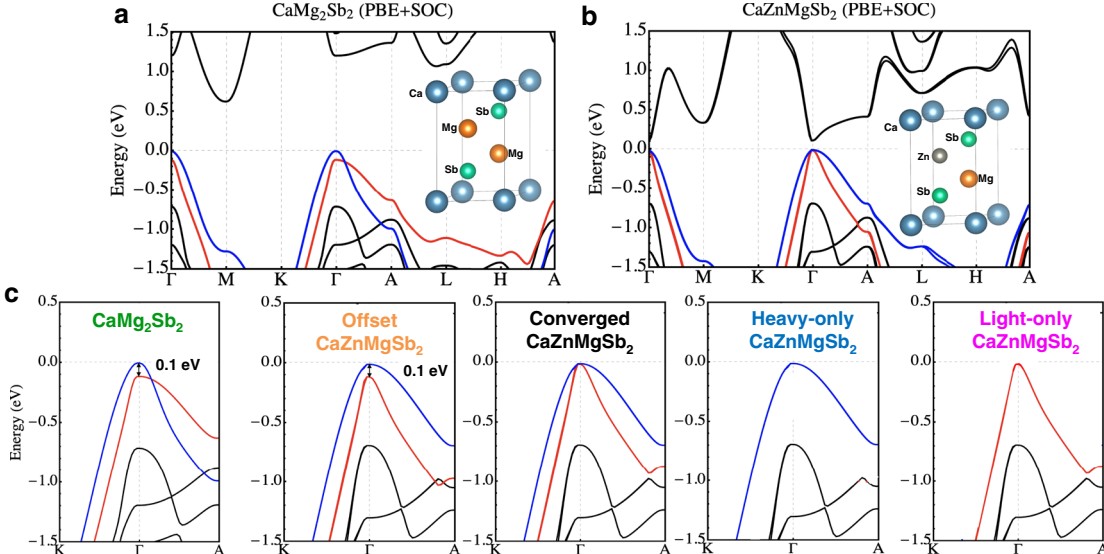

**Fig. 1 DFT band structures calculated with PBE including SOC. a** CaMg$_2$Sb$_2$ and (**b**) CaZnMgSb$_2$ band structures, with their crystal structure insets. Perfect ordering of Zn and Mg is assumed of the latter. Note that the two bands are essentially converged at Γ for Zn:Mg = 1:1. **c** The five band structure configurations (including hypothetical CaZnMgSb$_2$ configurations generated by manual adjustment of eigenvalues) subject to investigation via scattering and thermoelectric property computations.

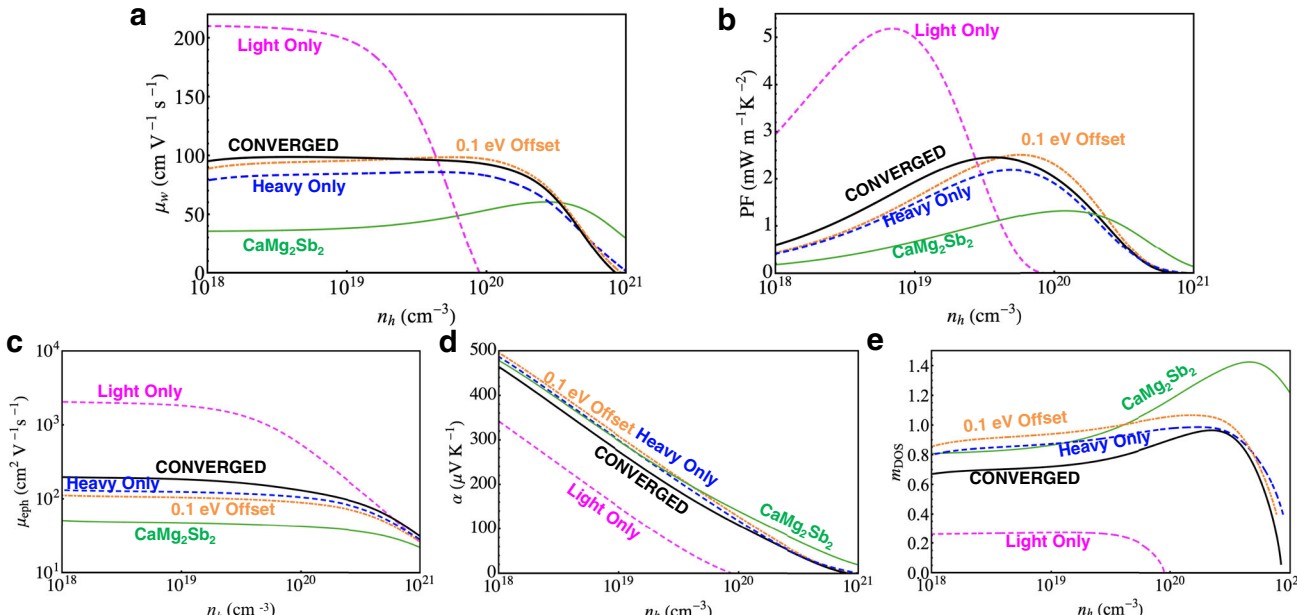

**Fig. 2 Calculated e-ph limited p-type properties of the band configurations depicted in Fig. 1c at 600 K. a** Weighted mobility, (**b**) the power factor, (**c**) mobility, (**d**) the Seebeck coefficient, and (**e**) the DOS effective mass.

shown in Fig. 2d, e. As a result, neither the PF nor $\mu_w$ benefits from closing the 0.1 eV offset. These behaviors visually summarized in Fig. 3 with experimental comparisons.

The responsible physics for the behavior described above lies with the altered scattering behaviors due to band convergence, as demonstrated in Fig. 4. First, interband scattering inherently limits $\mu$-enhancement. As the lower lighter band converges with the heavier upper band, overall scattering generally decreases for the lighter band but increases for the upper band (see Fig. 4a). The reason is that the lighter, lower band loses phase space as it is pushed towards the band edge whereas the upper band gains phase space for scattering as the lower band is introduced. The relative changes in the phase spaces in turn stems from the

changing local total DOS as the bands converge. We indeed find that the scattering rates largely scale as the total DOS as expected from DPS (see Fig. S3 in the SI). The overall $\mu$ still increases somewhat because the light lower band, which has higher group velocity, grows in population and lifetimes during convergence, contributing more to charge conduction. Nevertheless, in the presence of interband scattering, $\mu$ does not improve as much as it could in the absence of interband scattering. We note that in real samples, external mechanisms such as grain-boundary and disorder scattering further damage mobility[45–50]. These discussions, as well as calculated temperature-dependent transport properties compared with the experiment, are provided in Figs. S4 and S5 and Supplementary Discussion.

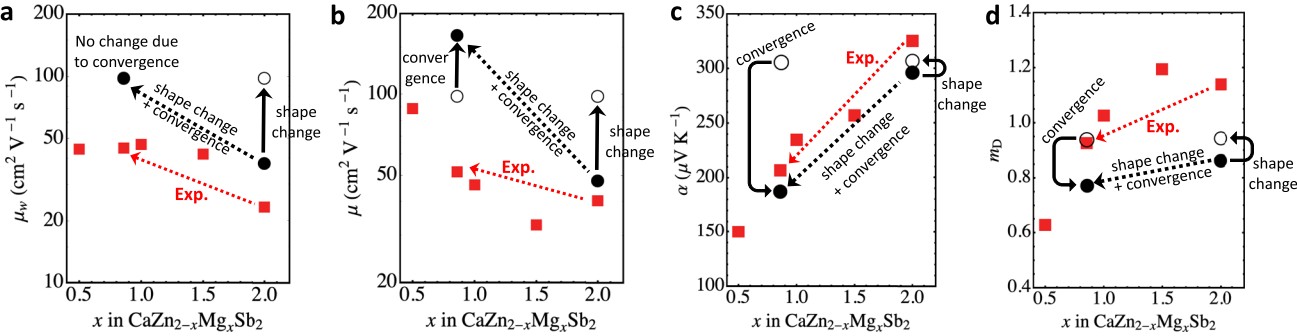

**Fig. 3 Computational explanation of the experimental trends with alloy composition. a** Weighted mobility, (**b**) mobility, (**c**) the Seebeck coefficient, and (**d**) the DOS effective mass. The temperature is 600 K. The experimental data points from ref. [37] are red squares. The calculated values are black circles. The two solid black circles correspond to $CaMg_2Sb_2$ at $x = 2$, and the band-converged $CaZnMgSb_2$, which we place at $x = 0.86$ (the experimental composition of band convergence). The open circle is $CaZnMgSb_2$ with 0.1 eV offset between the bands, which we place at both $x = 2$ and $x = 0.86$. The computational values track the experimental trends in two steps. First, the change from $CaMg_2Sb_2$ to the offset $CaZnMgSb_2$ is purely attributable to changes in band shapes. Then, the change from the offset $CaZnMgSb_2$ to the band-converged $CaZnMgSb_2$ is purely attributable to band convergence. Quantitative discrepancies between calculation and experiment are largely attributable to grain boundary scattering and disorder scattering; See Supplementary Fig. S4.

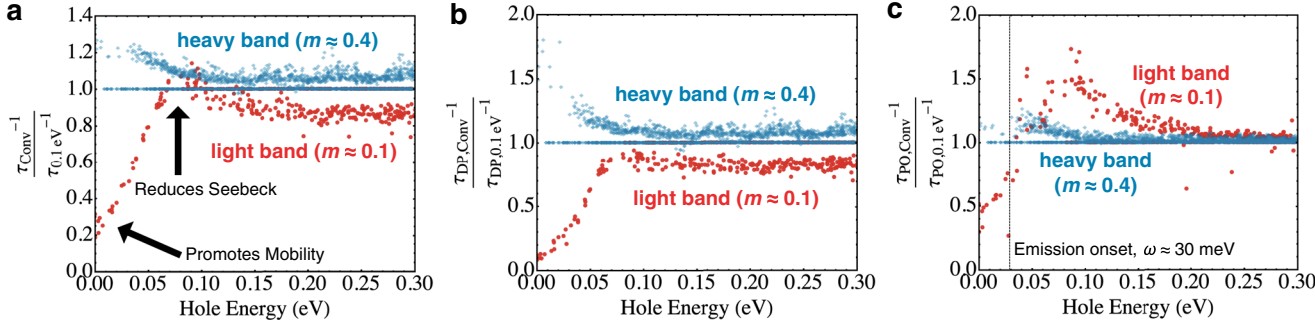

**Fig. 4 Scattering rate ratios (converged versus 0.1 eV offset) of $CaZnMgSb_2$ at 600 K.** A value above (below) unity indicates that scattering increases (decreases) due to convergence. The energy scale is zero-referenced to the VBM. **a** The overall scattering rate ratio. **b** The DPS ratio, whereby the upper band only increases in scattering, while the lower band only decreases in scattering. **c** The POS ratio, whereby both bands increase in scattering past the emission onset.

Second, the reduction of $\alpha$ can be traced to the portion of the lower band that increases in scattering due to convergence, at about 0.08 eV into the band in Fig. 4a. Increased scattering for the lower band due to convergence is not an allowed behavior under DPS as verified by Fig. 4b. We attribute this to POS as verified by Fig. 4c. POS is capable of increasing for both bands as they converge, past the emission onset. Because only the optical phonons near $\Gamma$ ($q \approx 0$) generate strong electric fields, POS characteristically intensifies with the proximity of the initial and the final states in the reciprocal space, as described by $\tau_{POS}^{-1} \propto |\mathbf{k} - \mathbf{k}'|^{-2}$[51,52]. Because energy surfaces of the two bands close on each other during convergence, POS increases for both bands, and about 0.08 eV into the light band, it overpowers DPS for increased overall scattering. For the light band states near the band edge that are below the emission onset, POS decreases just like DPS. Increasing (decreasing) scattering at high (low) energies reduces $\alpha$. Reduction of $\alpha$, in turn, results in a reduction of $m_D$ via Eq. (3). In all, these effects negate the minor improvement in $\mu$ leading to stagnation of $\mu_w$ and the PF.

We stress that the scattering behaviors observed in Fig. 4 are qualitatively reproducible using realistic model band structures and scattering models of ref. [53]. The reduction of $\alpha$ and $m_D$ due to band convergence with changing interband scattering is also reproducible by the same account. These more detailed explanations for the behavior of $\alpha$ and $m_D$ are given in the SI (see

Figs. S6–S8) and establish the generality of our findings for the Zintl alloys.

An underlying problem for these Zintl alloys is that the bands are converging at one k-point, allowing (1) zone-center acoustic phonons (near $\Gamma$) to incur interband DPS, and (2) interband POS to strongly manifest. Zone-center acoustic modes are some of the most populated. Zone-boundary phonons capable of coupling states across distant k-points in the Brillouin zone are of higher energies and less populated. Moreover, states located at distant k-points virtually cannot be coupled via POS owing to the relation $\tau_{POS}^{-1} \propto |\mathbf{k} - \mathbf{k}'|^{-2}$, whereas those located in the proximity of one another are easily coupled. Distinctively, as discussed above, interband POS reduces $\alpha$ because it preferentially scatters higher energy states of both bands during convergence. Of note, ionized-impurity scattering (IIS) similarly cannot couple distant states because $\tau_{IIS}^{-1} \propto \left( |\mathbf{k} - \mathbf{k}'|^2 + \gamma^2 \right)^{-2}$ where $\gamma$ is inverse screening distance[35,54]. Furthermore, in systems with inversion symmetry, it is known that interband DPS between symmetry-degenerate band pockets could be prohibited, for any phonon mode, by parity relations[55]. Interband scattering between two distinct bands at one k-point, though possibly prohibited for certain phonon modes by orbital symmetry, is generally allowed. All things considered, band convergence at one k-point is not as promising by design as that at distant k-points. Figure 5

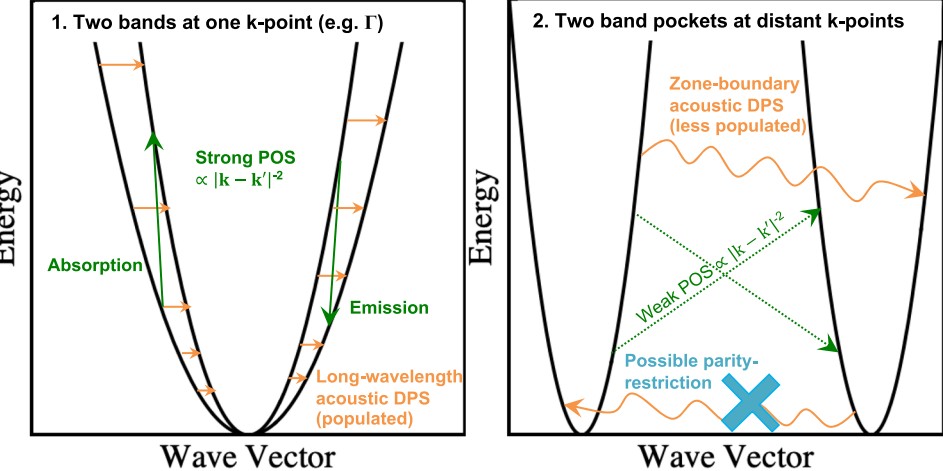

**Fig. 5 Schematic of e-ph interband scattering mechanisms.** They render intervalley scattering generally stronger between bands at one k-point than between bands at distant k-points.

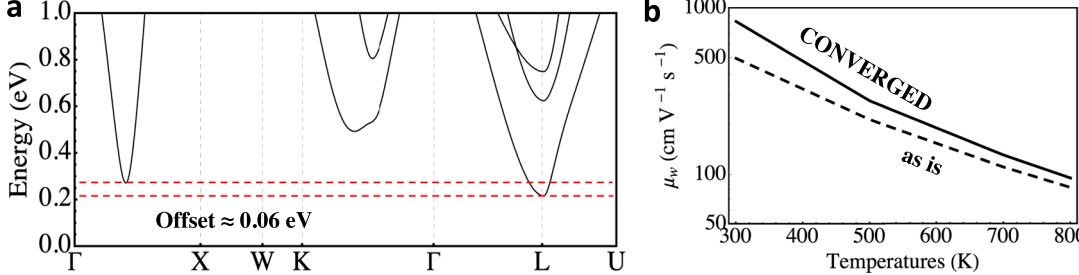

**Fig. 6 Full-Heusler $Sr_2SbAu$, an example of band convergence at distant k-points. a** The conduction band structure calculated with PBE including SOC. The dotted horizontal red lines indicate the minima of the $L$-pocket and the $\Gamma - X$ pocket. **b** The weighted mobility with and without band convergence.

summarizes inherent factors that render interband scattering of various kinds between states at faraway k-points generally weaker than those between states at one k-point.

To support the above hypothesis and draw a contrast to the Zintl scenario, we also perform a case study of $Sr_2SbAu$, a full-Heusler compound predicted to have high $n$-type performance[21] and in which two band pockets are nearly converged at distant k-points. The conduction bands of $Sr_2SbAu$ are depicted in Fig. 6a. The highly dispersive sixfold $\Gamma - X$-pocket has been found to be limited by POS and is minimally affected by intervalley scattering with the fourfold $L$-pocket[21]. The two pockets are originally offset by approximately 0.06 eV, with the heavier $L$-pocket being lower in energy. We take the same computational approach taken for the Zintl alloy and enforce full band convergence by pushing the $L$-pockets up by 0.06 eV. As expected from our discussion, when the bands converge, $\mu_w$ improves across all temperatures due to the valleys being distant in reciprocal space, as seen in Fig. 6b.

## Discussion

We stress that, experimentally, it is often difficult to quantify exactly how much benefit to performance is provided by band convergence alone because the effect of band convergence is never systematically isolated in experiments. Band convergence is typically achieved via alloying, whose process is expected to alter the band curvatures as well rather than purely achieving energy convergence. Such band-shape alteration occurs in $Mg_3Sb_{2-x}Bi_x$, which is found to be responsible for $\mu_w$ improvement in the alloys[56]. Likewise in this work, the band shapes change going from $CaMg_2Sb_2$ to $CaZnMgSb_2$. This, *not* band convergence, is responsible for the initial performance improvement in the

dilute-Zn regime coming off of the $CaMg_2Sb_2$-end, as can be read by comparing it to $CaZnMgSb_2$ with the same offset in Fig. 2. As $x$ decreases further, however, the two bands approach in energy but without improvement in $\mu_w$, as evidenced by both the experimental results in ref. [37] and our computational results. Furthermore, experimental reports on the benefits of band convergence were often established by means of the PF or $zT$ rather than $\mu_w$. The PF is a doping-dependent quantity unlike $\mu_w$[57], which may not have been optimized in each and every case, and $zT$ may simply benefit from concomitantly lowered $\kappa_{lat}$ in alloys, which is beside our point.

For clarification, our study does not mean that band convergence at one k-point can never be of benefit. If interband scattering is not as strong as what we find for $CaZn_{2-x}Mg_xSb_2$, band convergence even at one k-point could provide intrinsic benefit. However, our results and the underlying physics suggest that band convergence and multiplicity are much more likely to offer stronger benefits when convergence occurs at distant k-points. With hindsight, it is perhaps not a surprising corollary that hallmark cases of successful band convergence have generally occurred at distinct k-points. Filled skutterudite $Y_xCo_4Sb_{12}$ conduction bands have the twelvefold pocket along $\Gamma - N$ converging with the triply degenerate band minimum at the $\Gamma$-point for improved $n$-type performance[12,13]. In $p$-type $Bi_{2-x}Sb_xTe_3$ alloys, a threefold pocket along $\Gamma - Z$ and another along $Z - F$ converge[9,10]. In $p$-type PbTe, where band convergence was first demonstrated, two valence band pockets at the fourfold $L$-point and along the twelvefold $\Sigma$-line get close to convergence[6]. Moreover, it has recently been determined based on first-principles calculations that POS is the dominant mechanism for the PbTe valence bands, which would be weak between the

distant pockets[58,59]. A similar effect is observed in PbSe[8]. That said, $Mg_2Si_{1-x}Sn_x$ would warrant an in-depth study, whose alloys display higher PF than either end compound and undergo band convergence at one k-point at some intermediate $x$[15–17]. In Fig. S9 and Supplementary Discussions, we confirm that the bands do converge for the alloy by calculating its band structure with BandUP[60,61] and HSE06 hybrid functional[62,63], but discuss that it remains unclear whether it is indeed responsible for the peak performance at intermediate $x$.

In conclusion, band convergence does not always improve thermoelectric performance. We show theoretically that intrinsic electron-phonon scattering between bands is sufficient for rendering band convergence ineffective in Zintl $CaZn_{2-x}Mg_xSb_2$ alloys. Interband transitions are mediated by the highly populated zone-center acoustic phonons as well as long-wavelength polar optical phonons. During convergence, the former limits mobility enhancement by increasing scattering for the upper band while the latter lowers the Seebeck coefficient by simultaneously increasing scattering for the converging band. We find that these phenomena are by design much more likely if bands converge at the same k-point. Dispersive pockets at distant k-points tend to be less susceptible to strong intervalley scattering; such transitions require less populated zone-boundary phonons and virtually cannot be coupled by polar optical phonons. Therefore, convergence at distant k-points generally appears much more promising than that at one k-point by design. This study should offer an additional layer of guidance for experimentalists in their rational design and optimization of high-performing thermoelectrics. In particular, the weighted mobility relation $\mu_w = \mu N_v^{\frac{2}{3}} m_d$ may need to be reconsidered in a manner that accounts for the negative correlation between $\mu$, $m_d$, and $N_v$ via interband scattering.

## Methods

Electron-phonon scattering rates are derived from the imaginary part of electron self-energies, and ultimately calculated as

$$\tau_{\nu\mathbf{k}}^{-1} = \frac{2\pi}{N_\mathbf{q}} \sum_{\nu'\lambda\mathbf{q}} \left| g_{\nu'\nu\lambda\mathbf{k}\mathbf{q}} \right|^2 \Big[ (b(\omega_{\lambda\mathbf{q}},T) + f(E_{\nu'\mathbf{k}+\mathbf{q}},E_\mathrm{F},T))\delta(E_{\nu\mathbf{k}} + \omega_{\lambda\mathbf{q}} - E_{\nu'\mathbf{k}+\mathbf{q}}) \\ + (b(\omega_{\lambda\mathbf{q}},T) + 1 - f(E_{\nu'\mathbf{k}+\mathbf{q}},E_\mathrm{F},T))\delta(E_{\nu\mathbf{k}} - \omega_{\lambda\mathbf{q}} - E_{\nu'\mathbf{k}+\mathbf{q}}) \Big],\qquad(1)$$

where represents the e-ph interaction (scattering) matrix elements for coupling between electronic states $\nu'$ with wavevector k + q and $\nu$ with wavevector k due to phonon mode $\lambda$ of wavevector q and frequency $\omega$ Also, $b$ and $f$ are, respectively, the Bose–Einstein and Fermi–Dirac distributions for phonons and electrons, while $\delta$ is an energy-and-momentum-conserving delta function.

To compute Eq. (1), we use the EPW software[64–66], which interpolates coarse-mesh electronic states, phonon states, and e-ph interaction matrix elements onto dense k-point (for electrons) and q-point (for phonons) meshes using maximally localized Wannier functions[67–69]. The coarse-mesh matrix elements are calculated using density functional perturbation theory (DFPT)[70–72] as implemented in Quantum Espresso[73,74] using optimized norm-conserving Vanderbilt (ONCV) pseudopotentials[75–77] with the PBE functional. We calculate on coarse meshes of $8 \times 8 \times 4$ and $4 \times 4 \times 2$ and interpolate onto dense meshes of $80 \times 80 \times 60$ and $40 \times 40 \times 30$ for k and q, respectively. We calculate both deformation-potential and polar-optical scattering rates, whose matrix elements are given in Supplementary Methods.

Critically, we probe the effects of band convergence by manually shifting the interpolated eigenenergies of the bands prior to phase-space integration. We do so as to (1) force exact convergence at the Γ-point, (2) maintain the 0.1 eV energy offset (inherited from $CaMg_2Sb_2$), (3) simulate a hypothetical case where only the heavy upper band exists, and (4) simulate another hypothetical case where only the light lower band exists (see Fig. 1c for in the main text for the schematic). In strict terms, (3) and (4) actually involve offsetting the respective bands by full 1 eV, which essentially renders them irrelevant for thermoelectric transport while still maintaining the total number of electrons. These interventions thereby modulate only the $\delta$-functions in Eq. (1), offering a pure phase-space contrast between the fully band-converged and other cases, while keeping all other components identical, namely the band shapes.

Thermoelectric properties are computed by Boltzmann transport integrals using the BoltzTraP code[78], which we have modified in such a way that it takes band-and-k-dependent τ as inputs, in the same format as electronic eigenenergies, and then Fourier-interpolates them just as it does electronic eigenenergies. Weighted mobility is obtained from $\sigma$ and $\alpha$[57],

$$\mu_w = \frac{3\pi^2\sigma}{(2k_\mathrm{B}T)^{\frac{3}{2}}} \left[ \frac{\exp\left(\frac{|\alpha|}{k_\mathrm{B}} - 2\right)}{1 + \exp\left(-5\left(\frac{|\alpha|}{k_\mathrm{B}} - 1\right)\right)} + \frac{\frac{3|\alpha|}{\pi^2 k_\mathrm{B}}}{1 + \exp\left(5\left(\frac{|\alpha|}{k_\mathrm{B}} - 1\right)\right)} \right],\qquad(2)$$

by which account the DOS effective mass is

$$m_\mathrm{D} = \left( \frac{3\pi^2 n}{(2k_\mathrm{B}T)^{\frac{3}{2}}} \left[ \frac{\exp\left(\frac{|\alpha|}{k_\mathrm{B}} - 2\right)}{1 + \exp\left(-5\left(\frac{|\alpha|}{k_\mathrm{B}} - 1\right)\right)} + \frac{\frac{3|\alpha|}{\pi^2 k_\mathrm{B}}}{1 + \exp\left(5\left(\frac{|\alpha|}{k_\mathrm{B}} - 1\right)\right)} \right] \right)^{\frac{2}{3}},\qquad(3)$$

where $k_\mathrm{B}$ is Boltzmann's constant and $n$ is carrier concentration. These relations render $\mu_w$ and $m_\mathrm{D}$ doping-independent in the non-degenerate regime so long as scattering is also doping-independent[57]. For transport calculations, we use band gaps determined with the Tran and Blaha's modified Becke–Johnson (mBJ) potential[79,80], which are 1.12 eV $CaMg_2Sb_2$ and 0.33 eV for $CaZn_2Sb_2$. For the alloy, we use the Vegard's-law-interpolated value of 0.66 eV for the composition $CaZn_{1.14}Mg_{0.86}Sb_2$. For reference, the mBJ gap directly calculated for $CaZnMgSb_2$ is 0.80 eV. Supplementary Methods has further details.

While transport properties are calculated across a wide range of carrier concentrations ($10^{18} \sim 10^{21}$) cm$^{-3}$, we calculate and use $\tau^{-1}$ with the Fermi level fixed at 0.05 eV away from the VBM. In practice, e-ph scattering rates and lifetimes are usually not computed at every carrier concentration of interest because, unless doping is heavily degenerate, $\tau^{-1}$ has negligible dependence on the carrier occupation terms ($f$) in Eq. (1) as phonon occupation ($b$) is usually far larger (especially so for acoustic modes).

## Data availability

The data that support the findings of this study are available from the corresponding author upon reasonable request.

## Code availability

The custom codes used to perform the study are available from the corresponding author upon reasonable request.

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

## Acknowledgements

This work was intellectually led by the U.S. Department of Energy, Office of Basic Energy Sciences, Early Career Research Program, which funded J.P. and A.J. Lawrence Berkeley National Laboratory is funded by the Department of Energy under award DE-AC02-05CH11231. This work used resources of the National Energy Research Scientific Computing Center, a Department of Energy Office of Science User Facility supported by the Office of Science of the U.S. Department of Energy under Contract No. DE-AC02-05CH11231. G.J.S., M.D., and M.W. acknowledge NSF DMREF award #1729487.

## Author contributions

J.P. primarily designed the study, performed the computations, and led the manuscript writing under the supervision of A.J. who helped design the study and write the manuscript. M.D., Y.X., M.W., and G.J.S. helped design the study, interpret results, and draft the manuscript.

## Competing interests

The authors declare no competing interests.
