## [Peer Review File · Nature Communications]

REVIEWER COMMENTS

Reviewer #1 (Remarks to the Author):

Authors provide an impressive rethink of the band convergence in TE materials, since it has been difficult to experimentally measure the improvement in performance which was contributed by the band convergence. And it's of certain importance to quantify the additional scattering effect introduced by band convergence. The theoretical work and calculations are nice. I suppose it can be considered for publication in Nature Communications after minor revision. My suggestions are listed as follows.

1. Have the authors calculated other cases except those mentioned in the manuscript? Some more discussions about varied TE material systems can better support your conclusion.

2. The arrow that highlights the 0.096 eV offset should be located between the second and third of fig.1C.

3. Authors mention in the literature that you have used the BoltzTraP code to solve the BTE. As we know, BoltzTraP1 code uses the constant relaxation time approximation. If you are running the new BoltzTraP2 code in the manuscript, please also cite the more suitable reference: Computer Physics Communications 231(2018)140–145. Or if you use a code modified by your group, please provide a little more details.

Reviewer #2 (Remarks to the Author):

Band convergence is expected to be an efficient way to improve the electronic performance of thermoelectric materials. By performing a series of first-principles calculations and electron-phonon scattering analyses, the authors concluded that it is not always the case that band convergence leads to improved thermoelectric performance. Using CaMg₂Sb₂-CaZn₂Sb₂ Zintl system and full Heusler Sr₂SbAu as examples, the authors found that band convergence occurring at a single k point induces strong interband scattering and leads to weakened thermoelectric performance, and band convergence occurring at distant k-points preserves the single-band scattering behavior and leads to improved thermoelectric performance. The results presented are important for the design of high performance thermoelectric materials and the manuscript is well written. I recommend minor revision for the manuscript with the following comments:

(1) The authors concluded that, to improve the thermoelectric performance, "band convergence or multiplicity at distant k-points is more promising than it occurring at the same k-point." Then for typical thermoelectric materials, e.g., Bi₂Te₃, how to design to achieve alloy structures with band convergence occurring at distant k-points not at the same k-points?

(2) What is the reason that different bands have different η ranges in Figure 2, e.g., in Figure 2(a), "0.1 eV Offset" band and "CaMg₂Sb₂" band have different η range, even that "0.1 eV Offset" band has different η ranges in Figure 2(a) and Figure 2(c).

(3) It seems that Figure 3 and Figure 4 were wrongly exchanged? For example, the authors stated that "These behaviors visually summarized in Fig. 4 with experimental comparisons.", and "The responsible physics for the behavior described above lies with the altered scattering behaviors due to band convergence, as demonstrated in Fig. 3."

(4) There are some typos in the manuscript which should be corrected, e.g., "Eq. ??", "J. Am. Chem. Soc. 142:15464-15475 (2020)", and "Highly effective Mg₂Si_{1-x}Sn_x thermoelectrics".

Reviewer #3 (Remarks to the Author):

Band convergence has been a classical strategy for enhancing thermoelectric performance. In this study, the authors found that band convergence is not effective in the thermoelectric CaMg_2Sb_2 - CaZn_2Sb_2 alloys. When the band convergence occurs for bands at the same k point, the associated strong scattering of holes between different bands would offset the benefit from band convergence. It is suggested to accept this manuscript for publication after addressing the following concerns.

1. What is temperature that the calculated band structures correspond to in Fig. 1? Please note that the analyzed thermoelectric properties are at 600 K.
2. In Fig. 3, the experimental data are from samples with Na doping of $\text{Ca}_{0.09}\text{Na}_{0.01}\text{Zn}_{2-x}\text{Mg}_x\text{Sb}_2$ in the mentioned reference. It is necessary to make it clear in the figure caption.
3. According to the calculated band structures in Fig. 1, Zn alloying not only induces band convergence but decreases the band gap. The calculated results in Fig. 2 cover a wide carrier concentration range starting from a very low value. The conduction band might affect the thermoelectric properties significantly. Did the calculation consider the conduction band?
4. It is difficult to understand the DOS effective mass in Fig. 2e. Compared with the converged case, the 0.1 eV offset (artificial case) has the same band effective masses of heavy and light band. But the calculated DOS effective mass of 0.1 eV offset is higher.
5. In the fully converged case, is it necessary for holes to transmit between the heavy and light valence, resulting in interband scattering?

Reviewer #1 (Remarks to the Author):

Authors provide an impressive rethink of the band convergence in TE materials, since it has been difficult to experimentally measure the improvement in performance which was contributed by the band convergence. And it's of certain importance to quantify the additional scattering effect introduced by band convergence. The theoretical work and calculations are nice. I suppose it can be considered for publication in Nature Communications after minor revision. My suggestions are listed as follows.

→ Thank you for your favorable take of our work.

1. Have the authors calculated other cases except those mentioned in the manuscript? Some more discussions about varied TE material systems can better support your conclusion.

→ We were not able to do calculations on additional materials, but we have supplied an additional set of discussions on other band-convergence materials in light of our findings. We've added the following to our discussions:

Successful instances of band convergence in hallmark thermoelectric materials have all come from multi-pocket convergence in distant k-points. Skutterudite CoSb_3 conduction bands have the twelvefold pocket along $\Gamma - N$ converging with the triply degenerate band minimum at the Γ -point for improved n-type performance [12, 13]. In p-type $\text{Bi}_{2-x}\text{Sb}_x\text{Te}_3$ alloys, a sixfold pocket along $\Gamma - Z$ and another along $Z - F$ converge [9, 10].

[9] Kim, H.-S. et al. High thermoelectric performance in $(\text{Bi}_{0.25}\text{Sb}_{0.75})_2\text{Te}_3$ due to band convergence and improved by carrier concentration control. *Mater. Today* 20:452–459 (2017).

[10] Lee, K. H., Kim, S., Kim, H.-S., Kim, S. W. Band Convergence in Thermoelectric Materials: Theoretical Background and Consideration on Bi-Sb-Te Alloys. *ACS Appl. Energy Mater.* 3:2214–2223 (2020).

[12] Hanus, R. et al. Chemical Understanding of the Band Convergence in Thermoelectric CoSb_3 Skutterudites: Influence of Electron Population, Local Thermal Expansion, and Bonding Interactions. *Chem. Mater.* 29:1156–1164 (2017).

[13] Tang, Y. et al. Convergence of multi-valley bands as the electronic origin of high thermoelectric performance in CoSb_3 skutterudites. *Nat. Mater.* 14:1223–1228 (2015).

Please note that we have also supplied more detailed discussions of $\text{Mg}_2\text{Si}_{1-x}\text{Sn}_x$ for it is a notable case where the alloy with band convergence at one k-point seemingly delivers maximum performance. We confirm that band convergence does occur in that alloy system, but explain why it still remains unclear purely from experiments whether band convergence is the reason for the alloy's best performance.

2. The arrow that highlights the 0.096 eV offset should be located between the second and third of fig.1C.

→ We agree there needs to be better clarity of what our arrows are supposed to indicate. We addressed this by assigning vertical double-ended arrows to the first two figures that they have identical offsets of ~ 0.1 eV (we rounded up 0.096 eV).

3. Authors mention in the literature that you have used the BoltzTraP code to solve the BTE. As we know, BoltzTraP1 code uses the constant relaxation time approximation. If you are running the new BoltzTraP2 code in the manuscript, please also cite the more suitable reference: *Computer Physics Communications* 231(2018)140–145. Or if you use a code modified by your group, please provide a little more details.

→ We use modified BoltzTraP1 that intakes band-and-k-dependent lifetimes. It interpolates supplied band-and-k-dependent lifetimes equally as it does eigenenergies. Lifetimes can be supplied as inputs in the same format as eigenenergies. We have added the following to the Methods section:

Thermoelectric properties are computed by Boltzmann transport integrals using the BoltzTraP code, which we have modified in such a way that it takes band-and-k-dependent lifetimes as inputs, in the same format as electronic eigenenergies, and then Fourier-interpolates them just as it does electronic eigenenergies.

Reviewer #2 (Remarks to the Author):

Band convergence is expected to be an efficient way to improve the electronic performance of thermoelectric materials. By performing a series of first-principles calculations and electron-phonon scattering analyses, the authors concluded that it is not always the case that band convergence leads to improved thermoelectric performance. Using CaMg₂Sb₂-CaZn₂Sb₂ Zintl system and full Heusler Sr₂SbAu as examples, the authors found that band convergence occurring at a single k point induces strong interband scattering and leads to weakened thermoelectric performance, and band convergence occurring at distant k-points preserves the single-band scattering behavior and leads to improved thermoelectric performance. The results presented are important for the design of high performance thermoelectric materials and the manuscript is well written. I recommend minor revision for the manuscript with the following comments:

→ Thank you for your favorable take of our work.

(1) The authors concluded that, to improve the thermoelectric performance, “band convergence or multiplicity at distant k-points is more promising than it occurring at the same k-point.” Then for typical thermoelectric materials, e.g., Bi₂Te₃, how to design to achieve alloy structures with band convergence occurring at distant k-points not at the same k-points?

→ Though achieving multivalley convergence at distant k-points would likely depend on the specific system, a recent publication has also given some enlightenment from how and why multivalley convergence occurs in PbTe using tight-binding orbital interactions, which may be generalizable to some extent:

<https://dx.doi.org/10.1021/acs.chemmater.0c0374>

As for Bi₂Te₃ valence band convergence may occur at distant k-points when Sb is alloyed with Bi. Please refer to these papers, which have been referenced in the text:

<http://dx.doi.org/10.1016/j.mattod.2017.02.007>

<https://dx.doi.org/10.1021/acsaem.9b02131>

(2) What is the reason that different bands have different nh ranges in Figure 2, e.g., in Figure 2(a), “0.1 eV Offset” band and “CaMg₂Sb₂” band have different nh range, even that “0.1 eV Offset” band has different nh ranges in Figure 2(a) and Figure 2(c).

→ These have been fixed by extending the plots throughout the figure range.

(3) It seems that Figure 3 and Figure 4 were wrongly exchanged? For example, the authors stated that “These behaviors visually summarized in Fig. 4 with experimental comparisons.”, and “The responsible physics for the behavior described above lies with the altered scattering behaviors due to band convergence, as demonstrated in Fig. 3.”

→ Thank you for spotting this error. It has been fixed.

(4) There are some typos in the manuscript which should be corrected, e.g., “Eq. ??”, “J. Am. Chem. Soc. 142:15464{?15475 (2020)”, and “Highly effective Mg₂Si_{1-x}Sn_x thermoelectrics”.

→ Thank you for spotting these typos. They have been fixed.

Reviewer #3 (Remarks to the Author):

Band convergence has been a classical strategy for enhancing thermoelectric performance. In this study, the authors found that band convergence is not effective in the thermoelectric CaMg₂Sb₂-CaZn₂Sb₂ alloys. When the band convergence occurs for bands at the same k point, the associated strong scattering of holes between different bands would offset the benefit from band convergence. It is suggested to accept this manuscript for publication after addressing the following concerns.

→ Thank you for your favorable take of our work.

1. What is temperature that the calculated band structures correspond to in Fig. 1? Please note that the analyzed thermoelectric properties are at 600 K.

→ They are all 0 K DFT band structures. However, band gap corrections have been applied for transport calculations to properly capture temperature-dependent excitation (including bipolar effect).

2. In Fig. 3, the experimental data are from samples with Na doping of Ca_{0.09}Na_{0.01}Zn_{2-x}Mg_xSb₂ in the mentioned reference. It is necessary to make it clear in the figure caption.

→ We clarified this explicitly in Fig. 3 caption, as recommended.

3. According to the calculated band structures in Fig. 1, Zn alloying not only induces band convergence but decreases the band gap. The calculated results in Fig. 2 cover a wide carrier concentration range starting from a very low value. The conduction band might affect the thermoelectric properties significantly. Did the calculation consider the conduction band?

→ Yes the conduction bands are included in transport calculations. Bipolar effect would have been captured, but it is nearly nonexistent in the temperature ranges and the doping levels we study due to the fairly large band gap (> 0.6 eV)

4. It is difficult to understand the DOS effective mass in Fig. 2e. Compared with the converged case, the 0.1 eV offset (artificial case) has the same band effective masses of heavy and light band. But the calculated DOS effective mass of 0.1 eV offset is higher.

—> This is because they have different Seebeck, and hence the DOS mass is different between the two band structures owing to Eq. 2 in the Methods section. The DOS mass in the thermoelectrics context ties mobility to the weighted mobility, and they are experimentally inferred from either the Seebeck coefficient or weighted mobility. Therefore it is not quite a purely band-structure-dependent quantity as it is traditionally defined for parabolic bands: the geometric mean of directional curvature masses, which correctly yields carrier population for a given Fermi level.

5. In the fully converged case, is it necessary for holes to transmit between the heavy and light valence, resulting in interband scattering?

—> Interband scattering is a necessary component of our work, and it requires holes to scatter/transmit between the heavy and the light bands.

REVIEWERS' COMMENTS

Reviewer #1 (Remarks to the Author):

My concerns were fully addressed. I recommend the manuscript for publication.

Reviewer #2 (Remarks to the Author):

The authors have addressed all of my comments and I recommend it to be published in its current version.

Reviewer #3 (Remarks to the Author):

All the comments have been fully justified. I suggest to accept this paper for publication.